# Biophotons: New Experimental Data and Analysis

**DOI:** 10.3390/e25101431

**Published:** 2023-10-10

**Authors:** Maurizio Benfatto, Elisabetta Pace, Ivan Davoli, Roberto Francini, Fabio De Matteis, Alessandro Scordo, Alberto Clozza, Luca De Paolis, Catalina Curceanu, Paolo Grigolini

**Affiliations:** 1Laboratori Nazionali di Frascati, Istituto Nazionale di Fisica Nucleare, Via E. Fermi 40, 00044 Frascati, Italy; alessandro.scordo@lnf.infn.it (A.S.); alberto.clozza@lnf.infn.it (A.C.); luca.depaolis@lnf.infn.it (L.D.P.); 2Dipartimento di Fisica, Università di “Tor Vergata”, Via della Ricerca Scientifica, 00133 Rome, Italy; ivan.davoli@roma2.infn.it; 3Dipartimento di Ingegneria Industriale, Università di “Tor Vergata”, Via del Politecnico, 00133 Rome, Italy; roberto.francini@roma2.infn.it (R.F.); dematteis@roma2.infn.it (F.D.M.); 4Center for Nonlinear Science, University of North Texas, Denton, TX 76203, USA; paolo.grigolini@unt.edu

**Keywords:** biophotons, complexity, data analysis

## Abstract

Biophotons are an ultra-weak emission of photons in the visible energy range from living matter. In this work, we study the emission from germinating seeds using an experimental technique designed to detect light of extremely small intensity. The emission from lentil seeds and single bean was analyzed during the whole germination process in terms of the different spectral components through low pass filters and the different count distributions in the various stages of the germination process. Although the shape of the emission spectrum appears to be very similar in the two samples used in our experiment, our analysis can highlight the differences present in the two cases. In this way, it was possible to correlate the various types of emissions to the degree of development of the seed during germination.

## 1. Introduction

Nearly a hundred years ago, the Russian biologist A. Gurwitsch [1,2], doing experiments with onion plants by measuring their growth rate, observed that this was strongly influenced by the fact that the various seedlings were close or not and that this behavior remained even if the possibility of any bio-chemical exchange had been eliminated. On this basis, he hypothesized that plants emit a weak electromagnetic field capable of influencing cell growth. The scientific community completely forgot this interesting observation for many years, and only in the 1950s, Colli and Facchini [3,4], with the development of technology relating to radiation detectors, were able to make the first measurements of electromagnetic emissions coming from living organisms. This work was taken up again 30 years later by F.A. Popp [5] and co-workers who started extensive work to understand more in detail the origin and the meaning of such ultra-weak emission, hereby and after called bio-photons. Biophotons are an endogenous production of a very small flux of photons, of the order of 100 ph/sec, in the visible energy range characteristic of living organisms. This emission is completely different from the normal bioluminescence observed in some organisms because it is present in all living organisms, from plants to human beings, and it is several orders of magnitude weaker. Biophotons cannot come from the contribution of thermal radiation in the visible energy range because a simple calculation using the Plack distribution tells us that the intensity of this latter radiation is several orders of magnitude smaller than the biophotons contribution. Moreover, this emission ends when the organism dies; this excludes the possibility that it is the product of either some radiative decay produced by traces of radioactive substances present in the organism or by the passage of cosmic rays. The main characteristics of biophotons [5,6,7] are, besides the very small intensity, a practically flat emission within the energy range between 200 and 800 nm and the fact that any type of stress due, for example, to some chemical agents or excitation by light, induces a very fast increase in the emission up to several factors of magnitude followed by a relatively slow decrease to the normal values following a non-exponential law. For example, in delayed luminescence (DL) experiments, the return to normal emission occurs in a time that varies from a few tens of seconds to a few minutes [5,6,7,8,9,10,11]. This time scale is much faster than one of the typical germination processes, which normally takes place over tens of hours. This indicates that there are essentially two types of emission, one associated with the relaxation of molecular species excited due to the normal metabolic processes of the living organism and the other originating from the relaxation of excited states induced by the external stimulus [10,11]. Clearly, the two processes are closely connected, probably involving the same types of molecules, but, in our opinion, the decay channels responsible for spontaneous emission are different.

Despite the wealth of experimental results, the questions of what biophotons are, how they are generated, and how they are involved with life are still open. There are two hypotheses [5,6]. The first sees the emission as the random radiative decay of some molecules excited by metabolic events, while the second hypothesis assigns the emission to a coherent electromagnetic field generated within and between the cells by some biochemical reactions in which, perhaps, oxygen atoms are involved. At the same time, there is experimental evidence that such radiation carries important biological information [12,13]; for example, the radiation emitted by growing plants or organisms can increase by as much as 30% the cell division rate in similar organisms, the so-called mitogenetic effect [14,15,16].

Our group deals with bio-photon emission coming during the germination process of seeds of various kinds. The experimental setup we use to measure the biophoton emission is based on photomultiplier techniques, and it is constituted by a dark chamber and a photomultiplier sensitive to the visible energy range. The detector works as a photon counter, and the experimental data are the number of photons detected within a well-defined time window. In this way, our experimental data are essentially a time series where the counts detected in the chosen time window are reported as a function of the time calculated from the moment of closure of the experimental setup [17]. The duration of the experiment can vary from a few hours to many days, depending on the germination time of the considered seeds.

We have recently published a paper in which the time series generated by the biophotons emitted during the germination of lentil seeds has been analyzed using the diffusion entropy analysis (DEA) method. This method introduced in the literature in 2001 (see Ref. [18] for details) is based on the concept of complexity developed by Kolmogorov in the past. The Kolmogorov complexity is evaluated through a scaling index η, which is expected to depart from the ordinary value η = 0.5 if the signal shows some degree of anomalous complexity. The experimental time series is converted into a diffusional trajectory, and the complexity of the signal is derived through the evaluation of the Shannon entropy associated with the diffusional trajectory [17,18,19]. The main result of Ref. [17] is that the biophoton emission shows conditions of anomalous diffusion with a substantial deviation of the scaling coefficient from the ordinary value η = 0.5 throughout the duration of the experiment, which was 72 h. At the beginning of germination, the condition of anomalous diffusion is due to the presence of so-called crucial events, i.e., situations in which the system’s memory is reset to zero. As the seeds germinate and roots and leaves begin to develop, the type of complexity associated with the experimental data completely changes its nature, and the departure from the condition of random diffusion is due to the so-called fractional Brownian motion (FBM) [20] regime. This result is very similar to that found by the authors of Ref. [21], who analyzed the heartbeats of patients under the influence of autonomic neuropathy. In this case, the increasing severity of this disease has the effect of moving from a complexity condition generated by crucial events to a complexity condition characterized by the FBM infinite memory. Based on this analogy, the passage from a complexity characterized by crucial events to a condition dominated by FBM could indicate the passage from a normal physiological condition to a pathological one due to the lack of light in the experimental chamber, light necessary for the beginning of the chlorophyll photosynthesis process, with which plants generate the nutrients necessary for their growth. On the other hand, plants could have a completely different behavior during the growth process from that of humans since they do not have well-defined organs. Only at the beginning of germination is there a process of cellular differentiation, which leads to the development of leaves and roots, and this phase could require the presence of crucial events detected in the biophotonic emission. In other words, it cannot be excluded that the type of criticality inherent in the germination process requires a form of phase transition not yet known. It is interesting to note that Mancuso and collaborators [22,23] use the concept of swarm intelligence with reference to the network of roots generated by plants living in natural conditions. The presence of crucial events in the initial stage of germination could have something to do with the birth of this amazing radical intelligence.

In this paper, we present new experimental data related to the emission of a single bean and, at the same time, a new analysis of the various spectral components of the emission as a function of time. Furthermore, the distribution functions of the photocounts were analyzed in greater detail both for the lentil seeds and the single bean. All this was then related to the DEA analysis presented in the previous work.

## 2. Methods and Experimental Data

Our experimental setup was formed by a germination chamber and a photon counting system. Seeds were kept in a humid cotton bed and put on a Petri plate. The photon counting system consisted of a Hamamatsu (H12386 110) counting head placed on top of the germination chamber and an ARDUINO board driven by a PC with a Lab-View 11 program. The acquisition time window was fixed at 1 s. The germination chamber was built with black PVC to avoid any contamination of the light from outside. The whole system had a dark current of about 2 photons/sec at room temperature. Without any seeds or germination, there was a monotonic decrease in photon emission, which arrived in a few hours at the value of the electronic noise. This emission tail came from the residual luminescence of the materials, a consequence of the light exposure of the experimental chamber. Details can be found in Ref. [17].

All the analyses presented in this work were carried out with the help of the KaleidaGraph program version 5.02 [24].

The experiment was performed using 76 lentil seeds and a single bean seed. The results are shown in Figure 1. To clarify the comparison, the two curves have the same photon-counts scale, while the vertical black line in the panel of the single bean represents the time of 72 h, which is the duration of the entire acquisition time of the biophotons emitted by the lentils.

In both cases, the emission was activated by the watering process and analyzed in a wide time interval ranging from the end of the residual luminescence until the time when germination generated roots and leaves. Note that the time scales of the 76 lentils are completely different from the time scale of the single bean.

In Figure 2, we present the comparison between the two emissions. In order to highlight the common characteristics of the two emissions, we rescaled the time scale of the single bean by a factor of 0.164. In this way, it was possible to align the emission maxima of the two cases, the C peaks in the figure. The two curves have been moved further to have the zero of the time scale positioned in the first minimum. This means that the values 10 and 100, respectively, for the lentils and the single bean have been subtracted from the original time scale. To have the same number of counts in peak C, the values of the counts relating to the single bean were multiplied by a factor of 2.28.

This procedure was based on the use of the logistic equation [25] for the interpretation of the experimental data reported in Figure 1 and Figure 2. We hypothesized that the saturation time of the ordinary logistic equation in different systems corresponds to the maximum emission peak to a certain extent, and we rescaled the time scale of the bean so that its maximum photon emission rate coincided with that of the lentils. In particular, between 0 and 20 h, the lentil emission presented two peaks (B and C) separated by about 5 h; the same two peaks (B’ and C) were present in the emission of the single bean, but here they were separated by about 14 h. The biophoton emission of the bean showed a further peak (peak A) at about 43 h (these values in the bean time scale) after the minimum position at zero time in the scale. This peak was absent in the lentil emission. It should also be noted that in the germination phase, between zero and peak C, the growth phase of the emission presented at least two slopes.

The shape of the temporal evolution of the biophotons emission detected in our experiment seemed to be quite a general feature in the germinating phase of seeds, for example, the emissions of common wheat [26] (*Triticum aestivum*) and of seeds of *Arabidopsis thaliana* [27] are very similar to those presented here. This is quite interesting. The fact that the emissions of different seeds had a very similar temporal behavior led us to hypothesize the existence of a sort of generalized logistic equation as a universal property of the connection between system growth and photon emission.

The DEA analysis [17] and the behavior of the various spectral components (see Section 2.2 of this manuscript) indicate that the germination process had dynamics with time scales of the order of tens of hours, much longer than those typical of most experiments where living systems are subjected to external stimuli. Nevertheless, the emission as a function of time in this type of experiment (for example, the fact that the DL is practically independent of the frequency of the exciting light [5]) could indicate that the biophoton emission came from a unique process with different decay channels depending upon the type of the experiment.

By renormalizing the emissions by the number of seeds, we can also find the ratio between the number of photons emitted for each single seed of a different type. In this experiment, the ratio between the number of photons emitted by the lentils and the number of photons emitted by the single bean is about 133, a number very close to the ratio between the average weight of a single seed [28].

### 2.1. Data Analysis I—Probability Distribution Functions

In this section, we report the analysis of the emitted light in terms of the probability distribution function $P_{m}(T)$ of finding m counts in each acquisition time window T. In the semiclassical picture of the optical detection process, the phototube converts the continuous cycle-averaged classical intensity $\overset{-}{I}(t)$ in a series of discrete photocounts. Thus, the number m of photocount obtained in an integration time T is proportional to the intensity of the light that arrives on the detector [29].

A photocount experiment consists of a sufficiently large number of measurements of the number of photocounts in the same integration period T. The result of the measurement is expressed by the function $P_{m}(T)$ which represents the probability of obtaining m counts in the acquisition time T. The purpose of this measurement is to determine, if possible, the statistical properties of light through the properties of the distribution function, considering that, at least in some cases, there is a direct correspondence between the functional form of the $P_{m}(T)$ and the statistical properties we are looking for.

In a real experiment, it is practically impossible to repeat the same experiment many times, in our case where we have a system that is germinating and which, therefore, changes over time although with much longer times than acquisition time window T, 1 sec in our experiment. The distribution function is thus determined by means of a series of observations of a given duration, i.e., we detect the number of photons arriving in the phototube in one second and repeat this for the whole duration of the experiment, typically from one hour to the total time interval of the data set. The required photocounts distribution function is obtained as an average over successive starting time t of the function:(1)$$P_{m}\left(t,T\right)=\frac{{\left[\xi \bar{I}\left(t,T\right)T\right]}^{m}}{m!}\exp\left[-\xi \bar{I}\left(t,T\right)T\right]$$
where ξ is the detector efficiency and $\bar{I\;}(t,T)$ is the mean intensity of the light field on the phototube in the period from t to t+T [29]. So, $P_{m}\left(T\right)=\langle P_{m}(t,T)\rangle$ and the average is performed as previously described. On this basis, the mean number of counting is easily obtained as $\langle m\rangle =\sum_{m}m\;P_{m}(T)$, as well as the different moments and the variance of the distribution. It has been assumed that the emission is stationary. In our case, this is not strictly true, but this assumption becomes a good approximation for time intervals of the order of an hour or in the growth phase after the germination.

There are only some special cases where the average can be obtained in an analytical form. The simplest is that of a stable classic light wave where $\bar{I}\left(t,T\right)=\bar{I}$, i.e., the cycle-averaged intensity has a fixed value independent of the time [26]. In this case, the distribution $P_{m}(T)$ has a Poissonian form like
(2)$$P_{m}\left(T\right)=\frac{{\langle m\rangle }^{m}}{m!}\exp\left(-\langle m\rangle \right)$$
where $\langle m\rangle =\xi \bar{I}T$. A Poisson distribution is a sign of a system in a coherent state that corresponds to a classical electromagnetic wave [26,27], but, at the same time, this distribution also occurs for experiments where the integration time T is much longer than the characteristic time of the intensity fluctuations of the light beam. For the Poisson distribution, the variance is equal to the average $\sigma^{2}=\langle m\rangle$; any departure from the Poisson distribution is an indication of a non-classical nature of the light and can be measured by the Fano factor [23] F defined as $\sigma^{2}=F\;\langle m\rangle$.

The photocounts distribution can also be derived for a complete chaotic light [29], and it is equal to the photon distribution of a single-mode thermal source:(3)$$P_{m}\left(T\right)=\frac{{\langle m\rangle }^{m}}{{\left(1+\langle m\rangle \right)}^{1+m}}$$
This expression can be used for chaotic light of almost any type [29]. This formula can be generalized for thermal sources with M modes [30]:(4)$$P_{m}\left(T,M\right)=\frac{\left(m+M-1\right)!}{m!\left(M-1\right)!}{\left(1+\frac{M}{\langle m\rangle }\right)}^{-m}{\left(1+\frac{\langle m\rangle }{M}\right)}^{-M},$$
Thermal states are classical, and there is a relation:(5)$$\sigma^{2}=\langle m\rangle +\frac{{\langle m\rangle }^{2}}{M}$$
between the average number of counts and the variance. In general, the coefficient M can be very large; this means that the variance becomes almost equal to the average value, and we find the same relationship valid for the Poisson distribution. As a consequence, for very large M (greater than 20) [30], the thermal photocount distribution approaches the Poisson distribution. This implies that it is very difficult to discriminate between coherent and thermal states when many modes are present, which is in agreement with the discussion of Ref. [30].

The analysis of the dark counts, i.e., the counts measured with the black cap, has already been presented in Ref. [17] in detail. We remembered here that the experimental 〈m〉 value is consistent with the dark count data of this phototube [31]. We then proceeded with the emission analysis in the presence of seeds. In the reference [17], the count probability distribution functions $P_{m}\left(T\right)$ for the case of lentils have been derived using 1 h of emission at different germination times. The result of that analysis was to tend to super Poisson distributions in all cases, this being typical of chaotic or partially coherent sources.

In this paper, we present a similar analysis but using different sizes of the emission period, up to the entire measured data, and for the two types of seeds. In Figure 3, we show the comparison between the experimental $P_{m}(T)$ for the 76 lentils (panel a) and the single bean (panel b) with different types of fits. The measurement period used to obtain the count probability distribution function is from time 10 (hours) and time 83 (hours) to the end for the lentils and the single bean, respectively. See Figure 1 for clarity. In other words, we did not consider in both cases the first period where we could have contaminations due to the residual luminescence, and we waited for the first hints of rising in the counts that indicated (especially in the single bean) the beginning of the germination process. The experimental average value of the counts is $\langle m\rangle =23.5$ in the case of lentils, while the single bean gives a value $\langle m\rangle =7.9$, the variance is equal to 39.9 and 20.2 for lentils and the single bean, respectively. In both cases, this value is much bigger than the average count. The distribution relative to the single bean is clearly asymmetric, and it is impossible to obtain a good fit of it using either a Poisson (Equation (2)) or a many-mode functional form (Equation (4)). On the contrary, the distribution relative to lentils is optimally fitted with either one of the other two functional forms.

In both cases, the fit with a Poisson function is inaccurate, but this is not surprising considering the experimental difference between the mean values and variances and that the stationarity hypothesis is only weakly satisfied, having considered the entire time interval of measurement. The fact that the distribution of the 76 lentil seeds is strongly symmetrical and can be optimally fitted with a Gaussian is a clear indication that the various seeds have different germination times, which, therefore, give rise to emissions that are not in phase with each other. The use of a shorter time period for the calculation of the probability distribution function makes the hypothesis of stationarity more easily satisfied. In the case of single bean data, using a shorter emission period for the calculation of the function $P_{m}(T)$, we observe a transition to more symmetrical distributions. As an example, in Figure 4, we report the comparison between the experimental count probability distribution function (red squares) relative to the period from 200 (hours) to the end, with two fits using a Poissonian fit (solid green line) and a many-mode thermal function (points blue line).

In this case, the experimental average count is $\langle m\rangle =11.33,$ and the variance is $\sigma^{2}=12.93$, a value much closer to the average number of counts than that found in the previous case. The quality of the two fits is equivalent to producing a practically identical $\chi^{2}$ value [24]. In the Poissonian case, we obtain a value of the average counts equal to $\langle m\rangle =11.24\pm 0.05$, while the multi-mode thermal function gives 〈m〉=11.3±0.03 and M=48.0±0.03. It is interesting to note that in this last case, Equation (5) is almost satisfied. We performed this type of analysis for both the single bean and lentil seeds using emission periods ranging from one hour to several hours, up to the total, as shown previously. Some of this work is summarized in Table 1, where the distribution asymmetry S index is reported only for some of the time intervals chosen for the analysis. The asymmetry index S is defined as $S=\mu_{3}/\sigma^{3}$ where $\mu_{3}$ is the central moment of order 3, and σ is the standard deviation. A perfect symmetrical distribution has the value S=0. This analysis confirms that the count probability distribution functions related to the lentil seeds are much more symmetrical than those relative to the single bean in all the time intervals considered. This may be due either to a different characteristic of the seeds or also to the fact that in the case of lentils, we used many seeds to have a good signal/noise ratio.

When possible, typically for a short time window, the distributions can be fitted with either a Poissonian or a multi-mode thermal function. In any case, the experimental variance is always bigger than the mean value 〈m〉; this indicates a super-Poissonian type of behavior that is typical of either thermal emission or emission with a very short coherence time compared to the time window of the measurement. This makes it very difficult to discriminate between coherent and thermal states using the photo counting distribution analysis, which is in agreement with the discussion of Ref. [30].

### 2.2. Data Analysis II—The Different Spectral Components

The use of a turntable wheel holding a few long-pass glass color filters [32] makes possible an analysis in terms of the different spectral components of the emission. The wheel with the filters is placed between the germinating seeds and the detector and has eight positions. Six are used for the color filters, one is empty, and the last one is closed with a black cap [17]. The transmission coefficients of our filters and the efficiency of the phototube as a function of the wavelength of light are shown in Figure 5. The transmission coefficients are essentially theta functions positioned at the wavelengths written in the figure; thus, only light with wavelengths greater than the cutoff value shown in the figure can pass. The sensitivity of our phototube allows us to see the emission from near ultra-violet to yellow-orange with good sensitivity.

The total number of counts at time t without filters can be written as
(6)$$M_{tot}\left(t,T\right)=\int_{\lambda_{min}}^{\lambda_{max}}m\left(\lambda ,t,T\right)\;\alpha \left(\lambda \right)\;d\lambda ,$$
where m(λ,t,T) is the number of photons emitted from the sample at time t within the integration window of size T at a given wavelength, and α(λ) is the efficiency of the phototube. Inserting a filter with a transmission coefficient $f_{n}(\lambda )$ the number of counts $M_{n}(t,T)$ becomes
(7)$$M_{n}\left(t,T\right)=\int_{\lambda_{min}}^{\lambda_{max}}m\left(\lambda ,t,T\right){\;f}_{n}\left(\lambda \right)\;\alpha \left(\lambda \right)\;d\lambda ,$$
In Figure 6, we report the quantities $M_{n}(t,T)$ related to the different filters for both the lentils and the single bean. The count without any filters is also shown for comparison.

The different spectral components have a very similar shape to the emission without any filters. To see the possible different behavior of the various spectral components, we can do a monochromatizating calculation of the difference between the counts obtained using two filters with adjacent cutoffs. In this way, we have almost a monochromatic signal with an energy resolution of the order of 0.1%. The number of counts detected in the wavelength range defined by two filters with adjacent cutoffs is written as
(8)$$M_{n,s}\left(t,T\right)=\int_{\lambda_{min}}^{\lambda_{max}}m\left(\lambda ,t,T\right)\alpha (\lambda )\left[f_{n}\left(\lambda \right)-f_{s}(\lambda )\right]d\lambda ,$$
supposing now that the number of photons emitted from the sample in this wavelength window has a slight dependence on the wavelength, the average number of photons ${\bar{m}}_{n,s}(t,T)$ in each wavelength interval can easily derived as
(9)$$M_{n,s}(t,T)≅{\bar{m}}_{n,s}(t,T)\cdot \int_{\lambda_{min}}^{\lambda_{max}}\alpha \left(\lambda \right)\left[f_{n}\left(\lambda \right)-f_{s}\left(\lambda \right)\right]d\lambda ={\bar{m}}_{n,s}\left(t,T\right)\cdot I_{n,s},$$
(10)$${\bar{m}}_{n,s}\left(t,T\right)=\frac{M_{n,s}(t,T)}{I_{n,s}}$$

The value of the $I_{n,s}$ integral can be calculated numerically, and the average counts can be found using Equation (10). In Figure 7, we show the ratio between the different average counts ${\bar{m}}_{n,s}(t,T)$ and the total signal without filters for both lentils and the single bean. As usual, panel (a) refers to the lentil seeds, while panel (b) shows the result for the single bean. Because of the small counts in the case of a single bean, the ratio has been smoothed to clarify the behavior as a function of time. In the figure, the different letters indicate the position in time of the main emission peaks between 0 and the maximum, as used in Figure 2. In both cases, the different ratios are shown to change as a function of time. In other words, according to the moment of germination, the total signal is formed by different spectral components that change in relative intensity.

In the case of lentil seeds, the best signal/noise ratio allows us to say that the dominant components are those of orange (600–645 nm) and yellow-green (550–600 nm), in agreement with the results of Colli and Facchini [4]. It is interesting to observe how, while the high-energy components remain constant for the entire time of the measurement, the lower-energy parts clearly change in relative intensity throughout the duration of the measurement. In particular, the orange component is constant at first and then slightly decreases in the temporal region between 0 and peak B; this behavior is associated with a simultaneous increase in the yellow-green component of the spectrum. Here, it is the temporal region where germination begins and where the complexity is dominated by the presence of crucial events.

The behavior in the case of the single bean is not so clear because the signal-to-noise ratio is lower than in the lentil case. We can certainly say that in this case, the main components are orange and blue-green, while the yellow-green component becomes important in the last phase of germination, where the intensity of emission is at its maximum. It should also be noted that this component grows throughout the duration of the experiment, while the high-energy component (455–500 nm) grows considerably in the region between 0 and peak A and then remains constant in the region between this peak and peak B’. This may possibly be associated with the slope change in the growth of biophoton emission.

## 3. Conclusions and Suggestions for Future Works

In this work, we have analyzed in detail the emission of biophotons from lentil seeds and from a single bean throughout the germination period. We have highlighted the remarkable similarities in the form of emission, although the emission associated with the bean is characterized by the presence of an extra peak in the time period delimited by the 0 and the maximum of the emission (peak C); see Figure 2. Data were analyzed both in terms of the probability distribution functions of counts, using time windows of different sizes, and in terms of the different spectral components of the biophoton emission.

Although the analysis obtained using the probability distribution functions has several intrinsic difficulties in obtaining reliable information on the statistical properties of the emitted light, it is clear how this method should be used mainly for the emission coming from a single seed to avoid the problems related to the different times of germination of seeds and which significantly conditions the shape of this distribution, as shown in Figure 4 (lower panel).

The analysis of the various spectral components clearly shows how these change throughout the measurement period. There is a change in the relative weight of the different spectral compounds, which are related to the details of the emission, particularly the different slopes observed in the spectra. It is interesting to note how, in lentil emission, the orange component of the spectrum is clearly the dominant part of the spectrum up to about 30 h of emission. These first hours are also the temporal region where the orange and yellow-green components of the spectrum have an opposite trend. The first, which in any case remains the main part of the biophotonic emission, begins to decrease after an initial phase of practically constant value, while the second begins to grow to reach a value almost equal to that of the orange component. The region between 0 and 30 h (see Figure 7 in this paper) corresponds to the first three bands analyzed with the DEA method used in Ref. [17], and it is the region with the presence of crucial events. This behavior is a clear signal that during the germination period, the parts of the organism involved in the emission process change according to the degree of plant development. We can also hypothesize that the germination process, in this case, presents a kind of phase transition, highlighted by changes in the complexity patterns (crucial and non-crucial events) and by a different behavior of the spectral components because plants start to grow in an environment devoid of light and therefore chlorophyll synthesis cannot begin.

It is essential to increase the signal-to-noise ratio. Such improvement using this experimental setup can only be achieved by increasing the solid angle of acceptance of the detector. One possible way is to interpose Fresnel lenses between the sample and the phototube. Preliminary calculation obtained with a ray tracing type of analysis indicates a gain of about an order of magnitude. We are also verifying the possibility of using other types of methods to have a better signal/noise ratio. All this is, in our opinion, extremely important in order to be able to carry out measurements using single seeds and carry out a more precise spectral analysis.

## Figures and Tables

**Figure 1 entropy-25-01431-f001:**
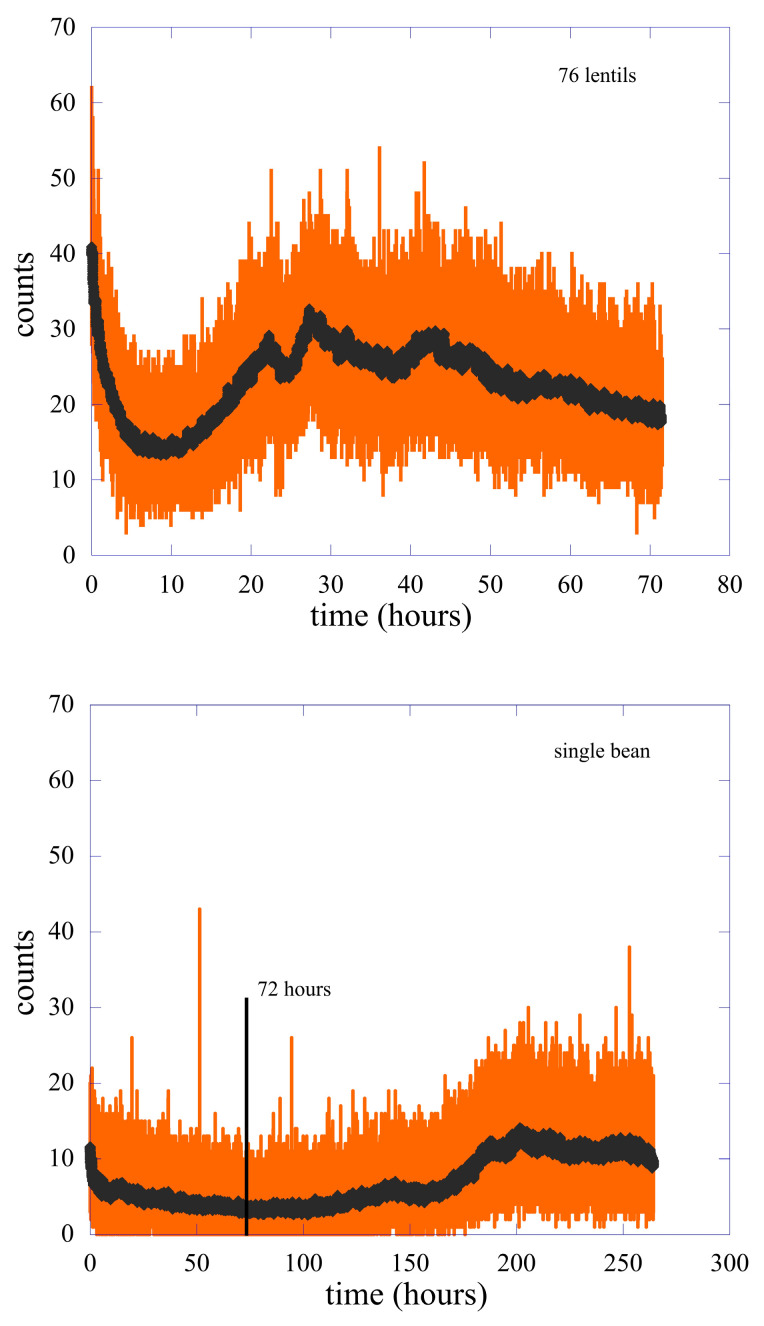
Biophoton emissions (counts per seconds, orange curves) of 76 lentil seeds. The vertical black line represents 72 h, which is the total acquisition time of the emission coming from the lentils. The black curves are the counts per second averaged over 1 min.

**Figure 2 entropy-25-01431-f002:**
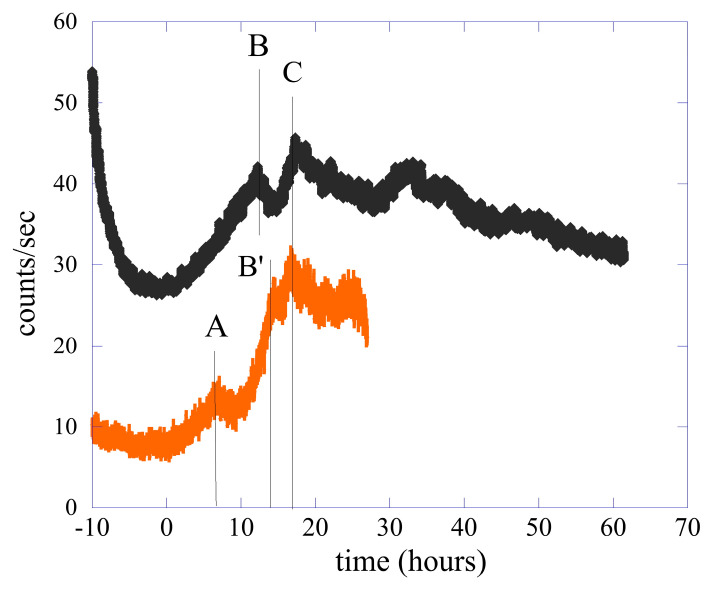
Comparison between the biophoton emission of the single bean with the emission of the 76 seeds of lentil. For clarity, the curve relating to the emission of lentils has been moved upwards, and it has been used in the time scale of the lentil’s emission. To obtain the time scale of the single bean, multiply the numbers by the factor 6.1 and add 100 h. The capital letters in the figure indicate the main emission peaks observed in the experimental data and are used in the discussion in the manuscript.

**Figure 3 entropy-25-01431-f003:**
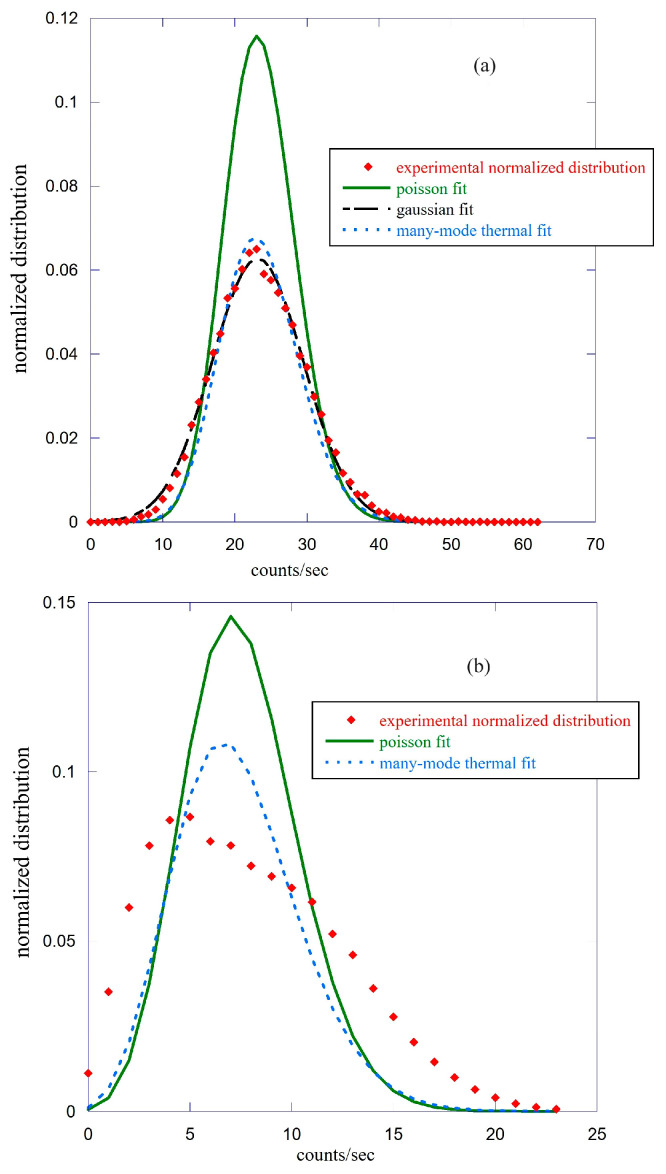
Comparison between the experimental count probability distribution function (red squares) of the lentil seeds (panel (**a**)) and the single bean (panel (**b**)) with three types of fits. The solid green line refers to a Poisson distribution function, the points blue line to the many-mode distribution function, and the points-dashed black line in panel (**a**) to a Gaussian distribution function.

**Figure 4 entropy-25-01431-f004:**
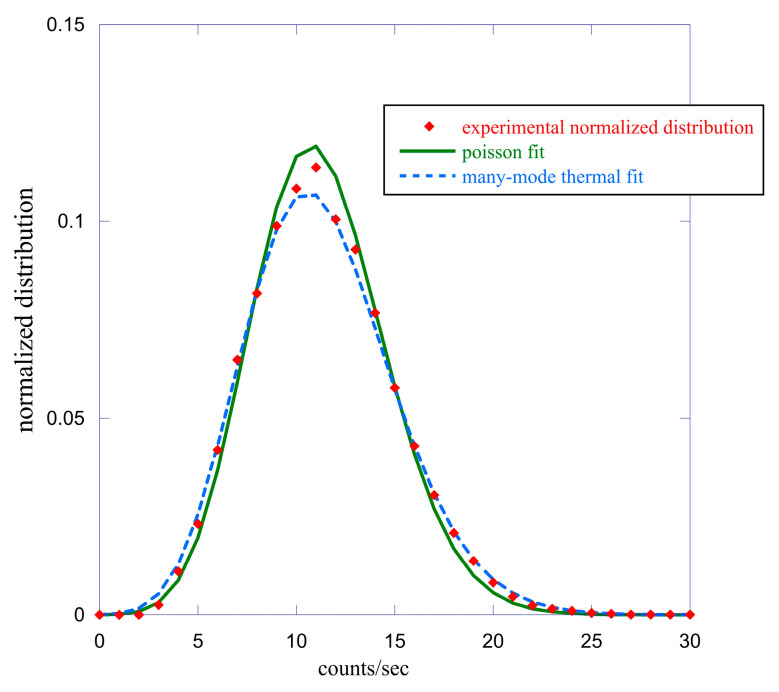
Comparison between the experimental count probability distribution function (red squares) relative to the single bean emission with two fits using a Poisson function (solid green line) and a many-mode thermal function (points blue line). Here, the emission period is between 200 (hours) and the end of the experiment.

**Figure 5 entropy-25-01431-f005:**
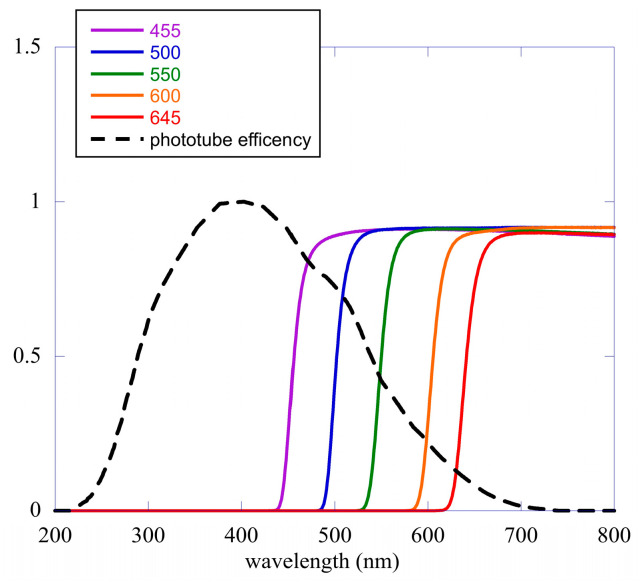
Transmission coefficients of the different filters used in our experiment and phototube efficiency. The wavelengths of the different cutoffs are reported in the figure. The dashed black curve is the efficiency of the phototube as a function of wavelength.

**Figure 6 entropy-25-01431-f006:**
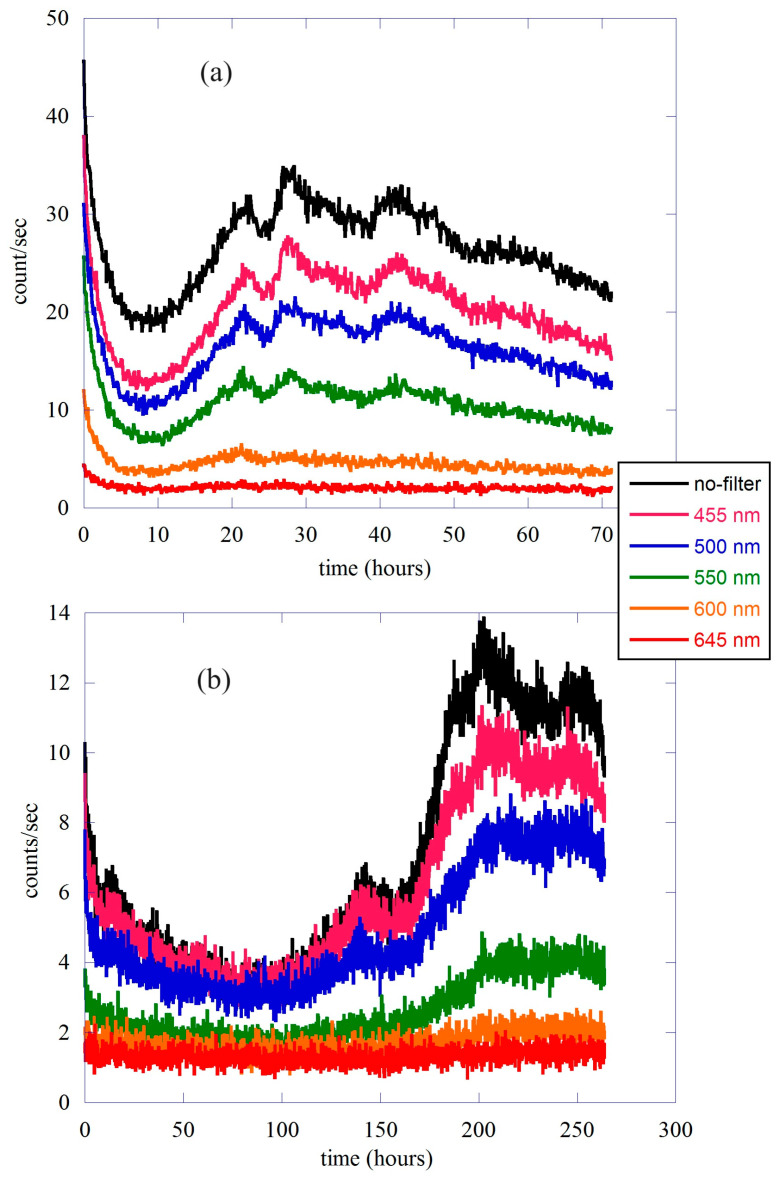
Counts per second averaged over 1 min for the different spectral components. Panel (**a**) is related to the lentil seeds, while panel (**b**) refers to the single bean.

**Figure 7 entropy-25-01431-f007:**
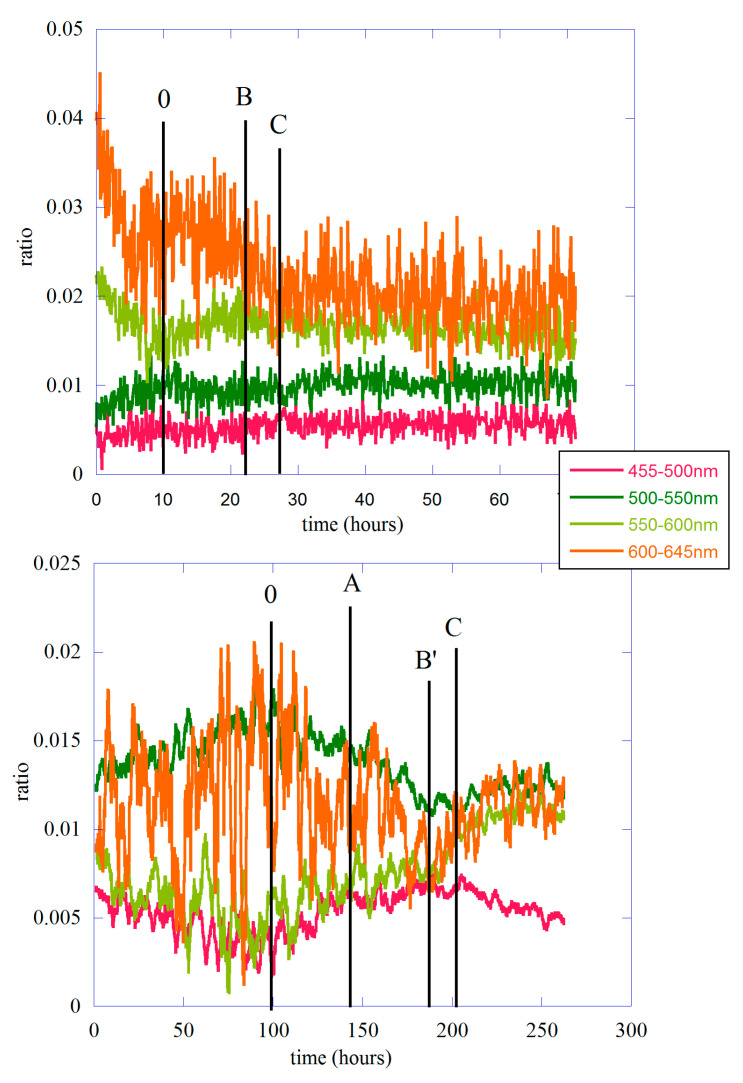
The ratio between average counts ${\bar{m}}_{n,s}(t,T)$ and the total signal. The different wavelength windows are reported in the figure with different colors. The capital letters in the figure correspond to those in Figure 2.

**Table 1 entropy-25-01431-t001:** S values for different time intervals for lentil seeds and the single bean. The time intervals are measured in hours. Refer to Figure 1 for details.

Single Bean		Lentil Seeds	
Time Interval	S Index	Time Interval	S Index
82–265	0.50	10–70	0.23
82–150	0.81	20–70	0.24
150–200	0.30	50–70	0.28
200–265	0.43	35–36	0.11

## Data Availability

All relevant data are available from the authors upon request.

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
