# Peer review of "Biophotons: New Experimental Data and Analysis"

_entropy, 2023, doi:10.3390/e25101431_

Round 1

Reviewer 1 Report

This manuscript comes up with a nice hypothesis of the following:

"the passage from a complexity characterized by crucial events to a condition dominated by FBM could indicate the passage from a normal physiological condition to a pathological one..."

and

"it cannot be excluded that the type of criticality inherent in the germination process requires a form of phase transition not yet known".

These are some very insightful projections of the mechanism and the associated process that may result from or result in biophoton emissions. 

Therefore the paper has merits to be published. In terms of the patterns of the biophoton emission that seemingly represent a simple exponential type of change on the photon count, the underlying systematic cause could be multi-fold. The authors may want to refer to some recent publications addressing possible underlying mechanism of the simple-to-complex time-course of the photon counts associated with biophoton emission in response to stimulation. 

However, the paper has some deficiencies. A major one is the scaling of the temporal indices between the two types of samples to draw a conclusion. That could be done after each type of samples are processed according to its own time-scale then normalizing to its total time-scale to present a pattern with respect to the relative time. The approach of stretching one time-scale to match with the time-scale of the other is problematic.

And there are quite a lot to improve regarding the presentation of a number of figures. The issues with figure presentations, among other issues scattered in the manuscript, are listed (incomplete, the authors shall extend the comments to the materials toward the latter part of the manuscript) in the following: 

Line 42  "alive organisms"? "living organisms"  

line 46, "visible" after range should be removed

line 47, "this latter" should be "the intensity of this latter radiation" 

line 67 to 68, obvious duplicative use of words

line 95, "which found" ---check grammar

Figure 1. It would be desirable to mark in b) the time window corresponding to the total time of a), and scale the photon-counts to the same scale. This will help compare the emissions from two kinds of samples.

Figure 1 and 2. Both need legend on the figure to mark which one is which (seed type).

Equation 1, if the detector efficiency is dimensionless, the mean intensity multiplied by time shall be dimensionless. Can you specify the units of the mean intensity?

line 224, "it results equal..." check grammar

Figure 3and 4 are poorly marked with difficult-to-distinguish solid-line styles. Please use line thickness and solid-dash-doted type to differentiate the lines. Not colors.

And legend is needed to mark the lines.

Figure 5--the bell-shaped line needs to be annotated as being the spectral response..

Need improvements. Some specific comments are shown above.

Author Response

Answers to Referee #1

This manuscript comes up with a nice hypothesis of the following:

"the passage from a complexity characterized by crucial events to a condition dominated by FBM could indicate the passage from a normal physiological condition to a pathological one..."

and

"it cannot be excluded that the type of criticality inherent in the germination process requires a form of phase transition not yet known".

These are some very insightful projections of the mechanism and the associated process that may result from or result in biophoton emissions.

We thank the referee for the consideration he shows of our work presented in this paper. The emission of biophotons is truly a very fascinating phenomenon but also very difficult to decipher as it presents some aspects that are truly difficult to understand, first the fact that it seems to be present in all living organisms, at least for what it is our knowledge.

Therefore the paper has merits to be published. In terms of the patterns of the biophoton emission that seemingly represent a simple exponential type of change on the photon count, the underlying systematic cause could be multi-fold. The authors may want to refer to some recent publications addressing possible underlying mechanism of the simple-to-complex time-course of the photon counts associated with biophoton emission in response to stimulation.

In this work we do not present any biophoton emission data from systems subjected to external stimuli. Only in the introduction do we make a very brief mention of this type of research without citing any references. We have corrected this point by inserting three specific references.

However, the paper has some deficiencies. A major one is the scaling of the temporal indices between the two types of samples to draw a conclusion. That could be done after each type of samples are processed according to its own time-scale then normalizing to its total time-scale to present a pattern with respect to the relative time. The approach of stretching one time-scale to match with the time-scale of the other is problematic.

Regarding the problem of rescaling the time scale of the single bean, we have used this procedure to highlight the fact that the emission of the two types of seeds seems to have a very similar trend. This is based on the popular logistics equation of different systems getting the maximum population at different times. We can make a comparison between them by reducing the time of the slower growth by a reducing factor, 0.164 in the case of the slower growth of the bean. We assumed that the saturation time of the ordinary logistics equation corresponds to some extent to the largest emission peak, and we rescaled the time scale of the bean to make its maximum rate of photon emission coincide with that of lentils. The result is surprising. As shown by Fig. 2 it is possible to hypothesize a sort of generalized logistic equation should exist as a universal property of the connection between system growth and photon emission. Clearly this is a hypothesis that should be verified through long systematic work on a large variety of seeds, but the results obtained by other authors (our references 23 and 24) seem to confirm this hypothesis. The results relating to the use of the logistic equation in the interpretation of experimental data will be presented in another work that we are preparing. Nonetheless, we have modified the text to reflect some of these considerations.

And there are quite a lot to improve regarding the presentation of a number of figures. The issues with figure presentations, among other issues scattered in the manuscript, are listed (incomplete, the authors shall extend the comments to the materials toward the latter part of the manuscript) in the following: 

We thank the referee for pointing out these errors and problems in understanding the figures. The text and figures have been corrected for these remarks.

Equation 1, if the detector efficiency is dimensionless, the mean intensity multiplied by time shall be dimensionless. Can you specify the units of the mean intensity?

The average intensity is the average number of photo counts that are measured in the time interval from t to t+T. Thus, the average intensity has the dimensions of the inverse of a time.

line 224, "it results equal..." check grammar

   correct

Figure 3and 4 are poorly marked with difficult-to-distinguish solid-line styles. Please use line thickness and solid-dash-doted type to differentiate the lines. Not colors.

And legend is needed to mark the lines.

 The figures have been changed in accordance with these suggestions, but we do not agree to put the legends inside the figures. In our opinion it is better that legends remains in the figures caption to avoid figures with too many signs, which become too confusing.

Figure 5--the bell-shaped line needs to be annotated as being the spectral response.

     A small sentence has been added in the caption of figure 5.

All changes are highlighted in red in the new version of the paper. Please note that the manuscript has also been modified to respond to the criticisms of the other referee.

Author Response

Answers to referee #2

Clarifying the biophysical characteristics and biological significance of biophoton emissions has always been a core scientific issue in this research field. In this manuscript, by conducting long-term dynamic detection of biophoton radiation patterns during the germination process of both population seeds (76 lentil seeds) and a single bean seed, the researchers employed a new statistical analysis algorithm [Diffusion Entropy Analysis (DEA)] for biophoton intensity and spectral analysis and achieved significant research results. Therefore, this study not only further validates the previous research findings, but also provides new technological analysis ideas and methods, which are very interesting for the study of biophotons. Main comments and suggestions are as follows:

We sincerely thank the referee for these words of appreciation for our work. Biophotons are a truly fascinating topic in our view that needs all possible work to clarify both their origin (they appear to be present in all living organisms) and whether they are even used to exchange information.

  1. Please provide product information on the manufacturer of the different filters used in the experiment and also the detailed transmission coefficients. The relevant data are only comparably shown in Figure 5;

The filters we have used are Edmund Optics long pass filters with these names: RG-645, R-60, OG-550, Y-50, GG-455. Data sheets can be found on the website of Edmund Optics. We have inserted this information in the text as a reference.

  1. The first and second parts of data analysis (data analysis I and data analysis II ) did not specify the software platform and specific analysis process used for data analysis;

All the analyses presented according to the brief theoretical notes contained in our paper were performed using KaleidaGraph software version 5.02. We have put a reference for this point. The DEA analysis (not present in this paper) has been performed with a homemade software.

  1. In this study, an important finding is that the biophoton radiation intensity exhibits relatively stable high-energy (short wavelength) components, but fluctuating low-energy (long wavelength) components (lines 385-392) during the germination process. Could authors provide a speculated explanation for underlying biological mechanism? Because it is obvious that such changes are not related to ambient light (observed experimentally in a dark box), but could be related to temperature and water molecules?

In the previous work (reference 14 in this paper) we have seen the evolution of the germination process of lentil seeds via the changes in patterns of complexity, from a statistic with crucial events to a one without them. The DEA analysis done in reference 14 has been performed dividing the entire measurement time into 6 regions and carrying out 6 DEA analysis for the emission belonging only to each of these time bands. The crucial events were detected in the first three temporal regions which correspond to the time interval between 0 and 30 (hours) in Figure 8 of this paper. These are also the temporal regions where the orange and yellow-green components of the spectrum have an opposite trend. The first, which in any case is the main part of the biophotonic emission, after an initial phase of practically constant value begins to decrease, while the second begins to grow to reach a value almost equal to that of the orange component. From this point on, i.e., in the time moment of transition to a statistic dominated by FBM, the two components have practically the same trend. On this basis it can be hypothesized that the lentil seed germination process is a process that presents phase transitions accompanied by changes in the complexity patterns (crucial and non-crucial events) and by a different behavior of the spectral components. Initially, due to the watering process, the seeds begin to use the nutrients present in the cotyledon to initiate germination and begin DNA replication. Subsequently, cell division takes place with the appearance of the root and the first leaves, essential for starting the synthesis of chlorophyll. Probably later the plant begins to die as it begins to grow in an environment totally devoid of light. We believe that all this cannot be linked either to the temperature, which is kept constant for the entire duration of the experiment, or to the water which comes into play at the beginning as a trigger of the germination process and then does not undergo any change for the entire measurement duration.

We have edited the conclusions to include part of this response.

All changes are highlighted in red in the new version of the paper, please note that the manuscript has also modified to respond to the criticisms of the other referee.  

Round 2

Reviewer 1 Report

Nice revision. Can be made better.

Figure 5 has the lines marked by the legend in the figure. Why can't you do the same thing for Fig. 3 and Fig. 4? Authors should not look-into the captions to know which line stands for what.

Figure 1 caption: there are duplicating words.

Figure 5: the curve of photo-detection efficiency shall be plotted in a line style that is visually explicitly different from the others. The line can be dashed, or much thicker, ,,,,

-----------

Whereas your argument is that there seems to have a common temporal-course underlying the spontaneous photon emission from the seeds, a potentially much more robust way to assess and access that temporal-course is to give the biophoton generation a perturbation. That is to introduce stress or external stimulation. The stress-induced temporal response shall reveal the commonality of the process of the unknown biophoton generation, with much less uncertainty.

This reviewer believes that a discussion relating to this aspect in either introduction or discussion, supported by more literatures, will significantly enhance the significance of the findings of this work.

Author Response

Nice revision. Can be made better.

We thank the referee for the effort he did to help us in improving our manuscript. Going into the details of the suggestions:

Figure 5 has the lines marked by the legend in the figure. Why can't you do the same thing for Fig. 3 and Fig. 4? Authors should not look-into the captions to know which line stands for what.

Figure 1 caption: there are duplicating words.

Figure 5: the curve of photo-detection efficiency shall be plotted in a line style that is visually explicitly different from the others. The line can be dashed, or much thicker..  

The manuscript has been edited according to these suggestions.

Whereas your argument is that there seems to have a common temporal-course underlying the spontaneous photon emission from the seeds, a potentially much more robust way to assess and access that temporal-course is to give the biophoton generation a perturbation. That is to introduce stress or external stimulation. The stress-induced temporal response shall reveal the commonality of the process of the unknown biophoton generation, with much less uncertainty.

This reviewer believes that a discussion relating to this aspect in either introduction or discussion, supported by more literatures, will significantly enhance the significance of the findings of this work.

The referee raises the important question of how biophoton emission changes due to external stimuli and whether the study of the temporal response induced by the perturbation can be better used to obtain information on the mechanisms underlying the generation of biophoton emission. Typically, the living system reacts to an external stimulus with an increase in biophotonic emission and, if the disturbance is not too intense to kill the living organism, the emission returns to the normal level following a curve of non-exponential decay with a time that can vary from a few seconds to a few hours, depending on the stimulus used in the experiment. For example, in delayed luminescence experiments, where the biophotonic emission is measured after the sample has been illuminated by a pulse of light (monochromatic or not) which has a duration ranging from a few milliseconds to a few seconds, the return to normal emission occurs in a time that varies from a few tens of seconds to a few minutes. Similar behaviour also occurs when the system is subjected to chemical stress using a highly diluted toxic agent.  The fact that the temporal decay of luminescence after an external stimulus can be approximated with a power law presumably indicates the presence of many different decay channels to which different kinetics are associated. This behaviour is typical of the relaxation processes that occur from non-equilibrium states towards equilibrium in complex systems.

Our experiment measures spontaneous emission during the germination process which occurs on much longer time scales, typically on the order of tens of hours. The data analysis done with the diffusion entropy method (our previous work) and the behaviour of the various spectral components indicate dynamics that have typical times of hours. In our opinion this indicates that there are essentially two types of emission, one associated with the relaxation of molecular species excited due to the normal metabolic processes of the living organism and the other originating from the relaxation of excited states induced by the external stimulus. Clearly the two processes are closely connected, probably involving the same types of molecules, but, in our opinion, the decay channels responsible for spontaneous emission are different.

The connection between these two types of emission is a topic of great interest and we agree with the referee that such a study would be important to understand in depth the mechanisms underlying biophotonic emission. All this requires a lot of experimental work using both techniques on the same type of sample and a lot of theoretical effort in data analysis. Currently our experimental apparatus is not designed to measure biophotonic emission in the presence of external stimuli of any nature and we also believe that a detailed discussion of this topic requires a dedicated manuscript and is therefore outside the scope of this work.

In any case, part of this discussion has been included in the introduction and in paragraph two. The references have been updated in this regard.

All corrections regarding this part are in blue in the new version of the manuscript.